# Embolisation Using Microvascular Plugs for Peripheral Applications: Technical Results and Mid-Term Outcomes

**DOI:** 10.3390/biomedicines11082172

**Published:** 2023-08-02

**Authors:** Rémy Mailli, Olivier Chevallier, Amin Mazit, Alexandre Malakhia, Nicolas Falvo, Romaric Loffroy

**Affiliations:** 1Department of Vascular and Interventional Radiology, Image-Guided Therapy Center, Francois-Mitterrand University Hospital, 14 Rue Paul Gaffarel, BP 77908, 21079 Dijon, France; mailliremy@outlook.fr (R.M.); olivier.chevallier@chu-dijon.fr (O.C.); amin.mazit@chu-dijon.fr (A.M.); malakdebezak@hotmail.fr (A.M.); nicolas.falvo@chu-dijon.fr (N.F.); 2ICMUB Laboratory, UMR CNRS 6302, University of Burgundy, 9 Avenue Alain Savary, 21000 Dijon, France

**Keywords:** embolisation, microvascular plug, peripheral applications, outcomes

## Abstract

The Micro Vascular Plug^®^ (MVP, Medtronic) is a mechanical embolic agent available in small sizes that allows for distal embolisation. The objective of this retrospective observational single-centre study was to assess MVP embolisation procedures performed at a university hospital. The 33 patients who underwent MVP embolisation in 2021 were included (mean age, 64; 24 males and 9 females). The primary endpoint was technical success, which was defined as a full first-attempt occlusion with one or more MVPs, as documented on the end-of-procedure angiogram. In all patients, 51 MVPs were used overall, with other embolic agents in 23 of the 33 cases (usually coils and/or glue); 22 of the 33 procedures were emergent for bleeding and 11 were planned for other indications. Of the three technical failures, two were due to an angled target artery configuration precluding microcatheterisation and one to failure of the device to release from its wire. The technical success rate was thus 90.9%. No patient experienced MVP migration or other major complications. Five patients had recurrent clinical symptoms; in four cases, the cause was collateral development, and in one case, the cause was incomplete initial embolisation. No instances of recanalisation were documented during the short follow-up of 12 months, for a 100% secondary clinical success rate. At our tertiary-level centre, the MVP was both effective and safe for peripheral applications. Interventional radiologists should be conversant with the techniques and indications of MVP embolisation.

## 1. Introduction

Since the first report of embolisation in 1972, interventional radiology has steadily gained importance as a therapeutic tool and field of technological innovation [1,2]. Embolisation done to treat vascular lesions or stop bleeding seeks to occlude one or more blood vessels [3]. The many available embolisation agents used are categorised as resorbable (e.g., gelatine and beads) or permanent [4]. The latter include particles (non-resorbable beads), liquids (glues and copolymers) and mechanical devices (coils and plugs) [5]. Each of them has its specific features in terms of size, delivery, occlusive power and cost. The choice of the best embolic agent according to the clinical scenario and the technical constraints remains at the discretion of the interventional radiologist.

The Micro Vascular Plug^®^ (MVP, Medtronic, Minneapolis, MN, USA) is a mechanical embolic device whose specific design features are suited for use in small-to-medium-calibre vessels [6,7]. The ovoid, self-expanding, nitinol skeleton is coated with polytetrafluoroethylene and attached by a screw to a push wire. Four sizes are available. Once released, the MVP can be repositioned in the target zone up to three times. MVPs can be used in many indications [8,9,10,11,12,13,14]. Since the first report of MVP embolisation in 2014, only about 30 articles on the use of this device for interventional radiology have been published, compared with hundreds for other embolic materials [11,15,16,17,18,19]. Additional recent data on MVP embolisation are therefore needed.

The objective of this study was to describe the technical success, clinical effectiveness and mid-term outcomes of MVP embolisation used alone or combined with other materials. We considered both scheduled and emergent endovascular procedures. The novelty of this work is the range of indications where the MVPs were used for peripheral applications.

## 2. Materials and Methods

### 2.1. Study Design and Patients

This single-centre retrospective observational cohort study was conducted at the Dijon University Hospital (Dijon, France) in patients managed from 1 January to 31 December 2021. The study was approved by our institutional review board, which waived the requirement for informed consent in compliance with French law on retrospective studies of de-identified health data. Patients with decision-making competency at the time of the procedure gave their informed consent to the procedure in writing. We searched the medical device traceability, medical device vigilance and interventional radiology department databases to identify patients who underwent MVP embolisation during the study period.

### 2.2. Data Collection

All the study data were collected by a single investigator (R.M.). For each patient, the following data were entered into standardised forms: age, sex, procedure, type of vascular lesion, whether MVP was used alone or in combination with other techniques (and in the latter case, the other techniques used), the MVP size and model, and whether the procedure was scheduled or emergent. Complications directly related to MVP use were recorded. Finally, follow-up data during the first year after the procedure were recorded.

### 2.3. Embolisation Technique

The indication for embolisation was based on the clinical evaluation, haemodynamic status and computed tomography (CT) findings (e.g., active bleeding or false aneurysm). CT was performed routinely before the procedure to measure the involved vessels and help to design the interventional strategy. The patient was then taken to the interventional radiology platform. All embolisation procedures were undertaken by radiologists with more than 5 years of experience in transcatheter embolisation.

The right common femoral artery was the most common approach. Ultrasound guidance was used to insert an introducer using the Seldinger technique. Digital subtraction angiography with power injection of the target vessel was performed. The appropriate MVP was chosen based on the target vessel size, as measured via CT and digital subtraction angiography. Four MVP sizes were available (Figure 1). The unconstrained outer diameters of MVP-3Q, MVP-5Q, MVP-7Q and MVP-9Q were 5.3, 6.5, 9.2 and 13 mm, respectively. Among the four MVP sizes, the MVP-3Q and MVP-5Q were delivered using microcatheters (0.021″ and 0.027″ internal diameter, respectively) and the MVP-7Q and MVP-9Q by diagnostic catheters (0.041″ and 0.043″ internal diameter); these devices can expand to 3, 5, 7 and 9 mm, respectively. At the completion of the procedure, angiography was performed to check for proper MVP positioning with complete occlusion of the downstream vascular bed. A mechanical femoral closure device was then placed. The microcatheter was positioned in the selected landing zone, which was chosen to be as straight as possible, without bends or angles, to allow for accurate microcatheter navigation and optimal MVP deployment against the target vessel wall. The catheter was flushed with saline to prevent intraluminal clot formation before MVP insertion. The MVP was then advanced through the catheter until its distal radiopaque marker reached the catheter tip. The push wire was aligned with the axis of the MVP to ensure a straight release zone. The catheter was pulled back to a fixed point on the push wire and the MVP was released via anti-clockwise push-wire rotation to detach the screw under fluoroscopic guidance. Finally, the push wire was removed. The catheter was used for an angiography to check the result.

### 2.4. Endpoint Definitions

Technical success was defined as complete occlusion of the target zone, as documented via angiography after the first attempt, using at least one MVP with or without other embolic agents.

Primary clinical success was defined as the resolution of the treated abnormality, haemodynamic stability, and/or full control of exteriorised bleeding after the release of at least one MVP with or without other embolic agents during the first embolisation procedure.

Secondary clinical success was defined as recurrent bleeding after initial clinical success, with the success of a second embolisation procedure with non-MVP embolic agents.

### 2.5. Statistical Analysis

This study was purely descriptive. Quantitative variables were described as mean ± SD if normally distributed and as median [interquartile range] otherwise. Categorical variables were described as number (%). Given the heterogeneity of indications for MVPs use, no group comparisons were performed for this study.

## 3. Results

### 3.1. Study Patients

Figure 2 is the patient flow chart. The follow-up period was one year for all patients.

Table 1 reports the main features of the 33 included patients, in whom 51 MVPs were used alone or combined with other embolic agents depending on the target vessel size. Five patients with liver cancer in whom the outcomes were chiefly dependent on the response to chemotherapy were excluded from the analysis of clinical success.

### 3.2. Indications

The 33 embolisation procedures were done for 14 different indications, 22 for nine emergent indications and 11 for five non-emergent indications (Table 2). All 22 emergent procedures were required to stop bleeding. Among the 11 scheduled procedures, three were done to redistribute the arterial blood flow during chemotherapy port catheter implantation in patients with liver cancer; three to manage arterio-venous fistula dysfunction in patients on chronic dialysis; one to treat a fortuitously discovered, post-stenosis aneurysm of the common hepatic artery; two to treat arteriovenous fistulas (one subcutaneous haemangioma and one retro-atrial mass); and two to allow for hepatic vascular mapping (Table 3).

The most often targeted artery was the gastroduodenal artery, with nine patients, including six who required emergency embolisation for severe bleeding and three patients who had chemotherapy port catheter implantation.

### 3.3. Technique

The procedures were performed by seven radiologists, each with at least five years of experience. The radiologist chose the size of the MVP during the procedure based on the target vessel diameter that was measured using the angiography images. MVP-3Q and MVP-5Q devices were released using 2.7 Fr microcatheters and the MVP-9Q device using a 5 Fr catheter. Overall, 51 MVPs were used for the 33 procedures. In 23 of the 33 procedures, additional embolic agents were used to increase the occlusion speed and effectiveness in patients that required emergent embolisation.

### 3.4. Technical Success Rate

MVP implantation was successful in 30 (90.9%) of the 33 patients. The three technical failures of MVP implantation were in patients #11, #16 and #17 (Table 2); in two cases, the procedure was technically successful after the implantation of embolic agents other than the MVP. In patient #11 with a ruptured aneurysm of the duodenopancreatic arcade, marked angulation of the target artery precluded catheterisation; Onyx™ and coils placed upstream and downstream of the angulation to exclude the target zone ensured clinical success. In patient #17 with a false aneurysm of the gastroduodenal artery, two MVPs that were used one after the other failed to release from the push wire; coils and glue ensured complete occlusion of the target zone. Finally, patient #16 with haemoptysis from a tumour was the only patient with technical failure of the procedure: MVP and glue embolisation were successful in occluding the right bronchial artery supplying the tumour but the second right bronchial artery originating from a common right-left trunk at the inferior aspect of the end of the aortic arch had a marked curve that precluded catheterisation. As a result, the bleeding recurred, requiring a second procedure, during which glue embolisation only partially occluded the common trunk and lefthand network but ensured lasting haemostasis. Figure 3, Figure 4, Figure 5, Figure 6 and Figure 7 illustrate the different technical and clinical scenarios.

### 3.5. Clinical Success Rate

The clinical success rate was assessed based on the 28 patients left after the exclusion of the five patients with liver cancer. Embolisation was successful with no recurrence during follow-up in 24 patients. In patient #10 with bleeding after partial right nephrectomy for a tumour and a false aneurysm of the superior polar artery, successful MVP embolisation was followed by recurrent bleeding caused by the development of collaterals, which were occluded using glue. In patients #19, #23 and #24, who had acute anaemia due to bleeding from a false aneurysm of the gastroduodenal artery with failed endoscopic treatment, initially successful embolisation combining MVPs, coils and glue was followed by recurrent bleeding due to collateral development; further embolisation with other agents and medical and surgical management ensured control of the bleeding.

### 3.6. Safety

No major complications related to the use of MVPs were recorded. In patient #20 with several false aneurysms after partial nephrectomy to remove a tumour, the use of six MVPs combined with coils and glue deprived 30% to 40% of the kidney parenchyma of its blood supply. This complication was expected. In patient #31 with an aneurysm of the common hepatic artery, a false aneurysm of the left humeral artery developed; pressure bandaging for 24 h ensured thrombosis of the lesion.

## 4. Discussion

In our cohort of 33 patients, including 32 who underwent distal embolisation with the two smaller MVP sizes, the technical success rate was 90.9%. In 20 (66.7%) of the technically successful procedures, one or more MVPs were used in combination with other agents. The two smallest sizes (MVP-3Q and MVP-5Q) were used in all the patients but one. These small devices were delivered using microcatheters, whose flexibility allowed for access to distal peripheral vessels. After the median follow-up period of one year, no patients had experienced recanalisation. Recurrent bleeding was noted in four of the patients with technically successful procedures and was consistently due to the development of collaterals, which were successfully treated via further embolisation. Of the three technical failures, two were due to sharp angulation of the target artery segment and one to failure of the device to release from its push-wire.

No major complications of MVP implantation were recorded in our study. Of our 30 technically successful MVP procedures, only ten were performed without additional embolic agents, which is a fact that may explain the absence of recanalisation. The additional embolic agents were designed to increase the speed and effectiveness of vessel occlusion, notably in patients that underwent emergent procedures for bleeding. No patient experienced MVP migration. Migration is best prevented by ensuring that the device is longer than the target segment [20,21,22]. Adding coils may decrease the risk of recanalisation [23].

MVPs are best suited for straight arterial segments. Curves, angles or tortuosities may preclude the insertion of the microcatheter required to deliver the MVP. Microcatheterisation failure due to angulation of the target artery occurred in two of our patients.

Of the five patients who experienced recurrent bleeding requiring a second embolisation procedure, one had had a partial technical failure due to one of the target arteries being too markedly curved to allow for catheterisation. In the other four patients, the recurrent bleeding was due to the development of collaterals.

Advantages of the MVP include the ability to reposition the device up to three times and the very limited CT artefacts. The risk of migration is low compared with particles and glues. The Amplatzer Vascular Plug requires a catheter size of 4 Fr, allowing only proximal embolisation. The two smallest MVP sizes, in contrast, can be delivered through microcatheters advanced into distal vessels. A single MVP was sufficient in 20 of the 30 technically successful procedures in our study, resulting in shorter procedures compared with the use of coils. In our patients, the main reasons for choosing the MVP were the need to achieve prompt haemostasis in patients with haemodynamic instability and bleeding from distal arteries that were too small to allow for the implantation of larger mechanical devices. Compared with coils, which can also be implanted in small arteries, the MVP provides faster haemostasis.

The main limitation of our study was the retrospective observational cohort design. The patients were managed at a tertiary care centre by highly experienced radiologists, and the high technical success rate may not be generalisable to other centres. The follow-up time was short and the sample size was small. Other embolic agents were used in addition to MVPs in two-thirds of cases, hampering an assessment of the effectiveness of the MVP itself.

## 5. Conclusions

Distal MVP embolisation, usually in combination with other embolic agents, was effective in our study. There were no instances of MVP migration or other major complications. The best indications for using MVP to occlude arteries may be distal arterial bleeding with haemodynamic instability requiring faster haemostasis than expected with coils alone. The MVP can be used in combination with other materials to expedite haemostasis. Interventional radiologists should be conversant with the indications and use of the MVP.

The novelty of this study lies in the potential range of indications where the MVPs can be used for peripheral applications. It highlights the fact that a certain time is needed to obtain complete occlusion after system delivery in order to avoid the use of other embolic materials. However, MVPs may also be used in combination with several embolic agents with good results. The versatility of MVPs makes having them in the therapeutic arsenal of utmost importance for different potential clinical scenarios.

## Figures and Tables

**Figure 1 biomedicines-11-02172-f001:**
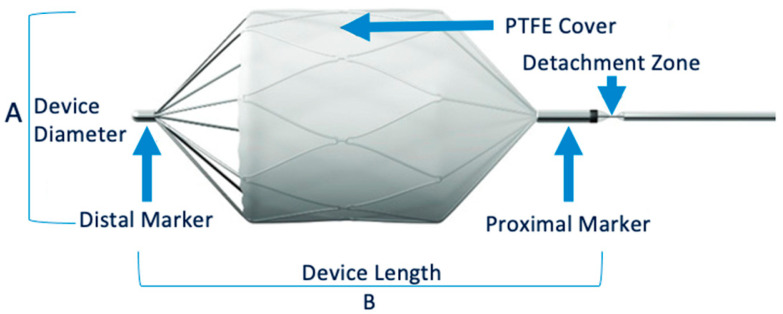
MVP^®^ Micro Vascular Plug System.

**Figure 2 biomedicines-11-02172-f002:**
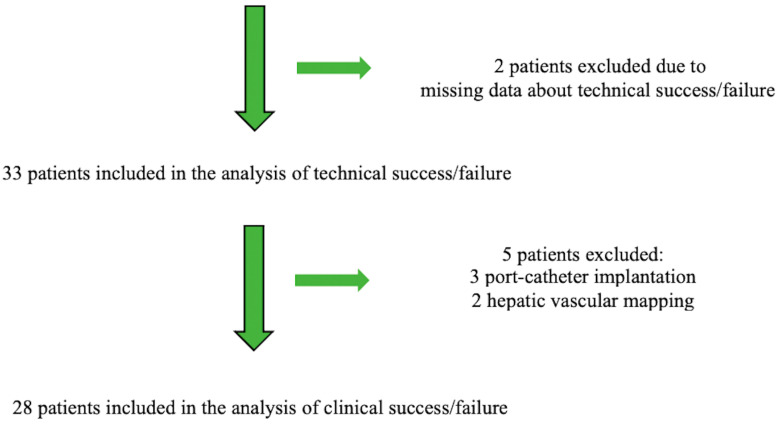
Patient flow chart.

**Figure 3 biomedicines-11-02172-f003:**
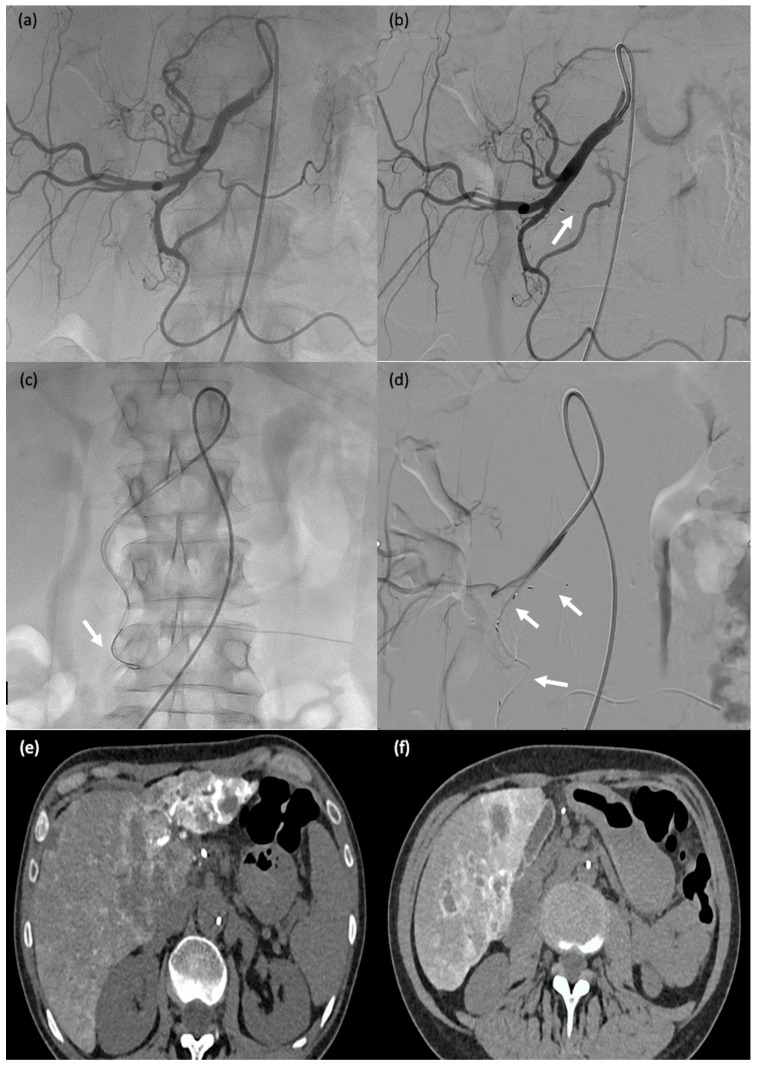
A 60-year-old man (#1) with diffuse metastatic disease from colonic cancer that was unresponsive to chemotherapy and required intensification via chemotherapy port catheter implantation. (**a**,**b**) MVP-3Q implantation (arrow) to occlude the left gastric artery before insertion of the chemotherapy catheter. (**c**,**d**) Catheter insertion in the gastroduodenal artery, occlusion of the gastroduodenal artery and stabilisation of the chemotherapy catheter via implantation of 2 MVP-3Qs through a smaller microcatheter inserted in the side-whole of the chemotherapy catheter (arrows). (**e**,**f**) Computed tomography after arterial contrast medium administration showing excellent enhancement of all metastatic lesions in both liver lobes.

**Figure 4 biomedicines-11-02172-f004:**
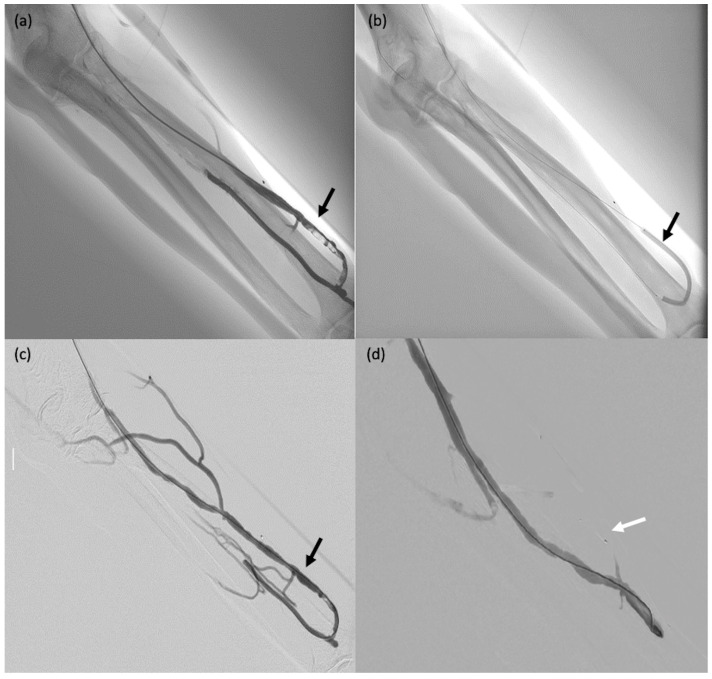
A 75-year-old male patient (#29) on dialysis for chronic kidney disease with dysfunction of the arteriovenous fistula. (**a**) Post-anastomosis venous thrombosis (arrow): the patient was taken to the operating room for thrombectomy. (**b**,**c**) Thromboaspiration and angioplasty (arrows). (**d**) MVP-5Q implantation to occlude collateral venous circulation, thereby redistributing the blood flow (arrow).

**Figure 5 biomedicines-11-02172-f005:**
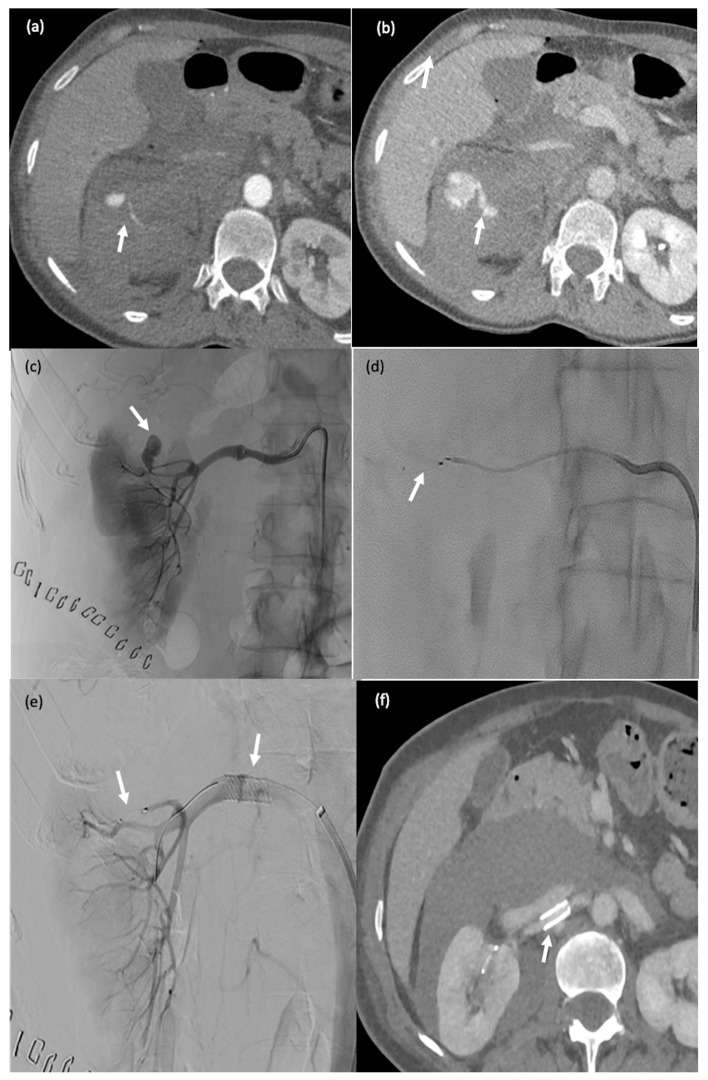
This 65-year-old female patient (#10) started experiencing severe lumbar pain six days after partial nephrectomy for renal cancer with renal artery clamping. Her haemoglobin level dropped by 3 g/dL and her haemodynamics were unstable. Active postoperative bleeding was suspected, and she was taken to the computed tomography suite. (**a**,**b**) Active retroperitoneal bleeding (false aneurysm) (arrow). (**c**–**e**) Angiography showing a false aneurysm of the superior polar artery of the right kidney ((**c**), arrow), which was embolised using an MPV-3Q ((**d**), arrow); stenting of the renal artery (clamp injury) ((**e**) arrow). (**f**) CT two days after embolisation: there was no active bleeding and the MVP-3Q and stent were properly positioned (arrow).

**Figure 6 biomedicines-11-02172-f006:**
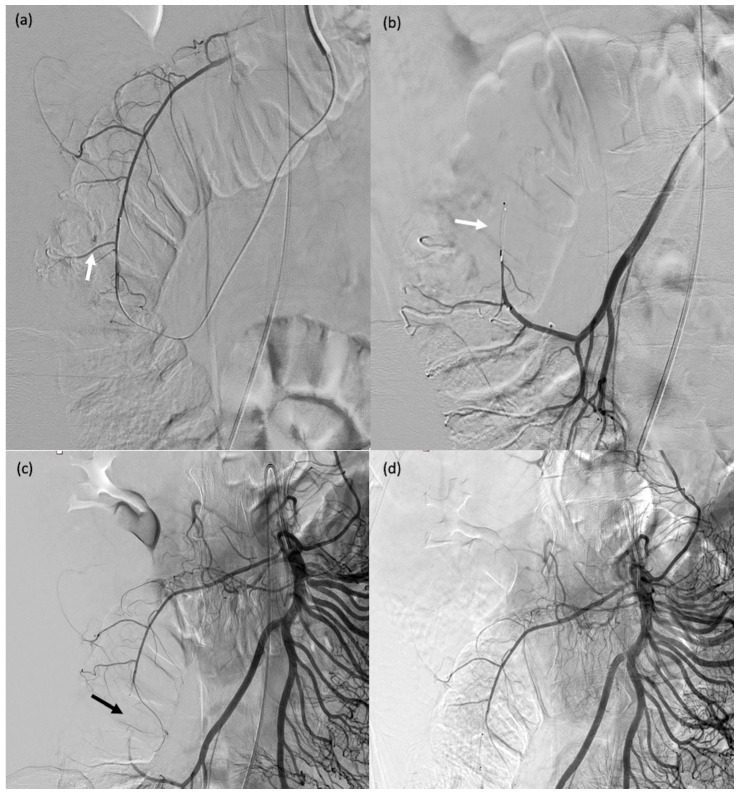
A 32-year-old female patient (#13) who passed blood per rectum 2 days after a right partial nephrectomy; computed tomography showed active bleeding from the right colon; due to haemodynamic instability, she was transferred to the operating room for emergent angiography. (**a**) Angiography showing contrast agent extravasation at the right colon (arrow). (**b**–**d**) Catheterisation of the target arterial branch was not feasible and a single MVP-3Q was therefore used for sandwich embolisation of the right arterial trunk at the appropriate level (arrows).

**Figure 7 biomedicines-11-02172-f007:**
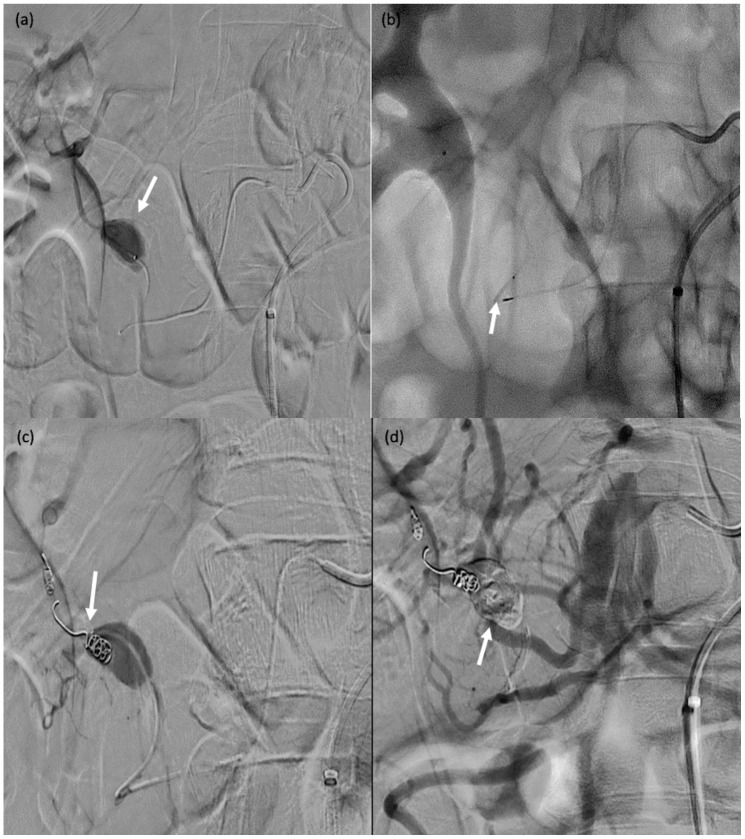
A 71-year-old male (patient #11) with abdominal pain and haemodynamic instability. (**a**) Rupture of a false aneurysm of the posterior pancreaticoduodenal arcade (arrow). (**b**) Failure of initial catheterisation for MVP-3Q implantation due to angulation of the target artery (arrow). (**c**) Coil embolisation of the downstream branch (arrow). (**d**) Onyx™ embolisation of the aneurysmal sac and upstream branch (arrow).

**Table 1 biomedicines-11-02172-t001:** Main features of the 33 included patients.

Variables	N or Mean
Age, years, mean (range)	64 (32–85)
Males/females	24/9
Number of procedures	33
Number of MVPs used	51
Reason for embolisation, No. of patients	
Bleeding	22
Other	11
Approach, No. of patients	
Arterial	29
Venous	4
Degree of urgency	
Emergent procedure	22
Scheduled procedure	11

**Table 2 biomedicines-11-02172-t002:** Indications, target vessels, MVP sizes and other embolic agents used.

Patient	Age/Sex	Indication	Target Vessel for MVP	MVP Size	Number of MVPs	Other Embolic Agents
1	60/M	Chemo. port catheter	Pyloric/gastroduodenal	3	4	None
2	62/M	CKD, AV fistula dysfunction	Branch of left cephalic vein	3	1	Coils
3	71/M	Hepatic vascular mapping	Left hepatic artery	3	1	Coils + glue
4	51/M	False aneurysm	Pancreaticoduodenal arcade	3	2	None
5	82/M	False aneurysm	Left superior gluteal artery			None
6	54/M	False aneurysm	Arterial branch feeding pancreatic pseudocyst			Coils + glue
7	62/F	Bleeding duodenal bulb ulcer	Gastroduodenal artery	3	2	Coils + glue
8	63/M	Haemoptysis	Right intercostobronchial artery	3	2	Glue
9	63/M	Retro-atrial mass: AVM	Branch of external carotid artery–left external jugular vein	3	2	Coils + Onyx™
10 ^b^	65/F	False aneurysm	Superior polar artery of right kidney	3	1	None
11 ^a^	71/M	False aneurysm	Pancreaticoduodenal arcade	3	1	Coils + Onyx™
12	76/M	False aneurysm	Left superior gluteal artery	3	1	Coils
13	43/M	Hepatic vascular mapping	Left hepatic artery	3	1	Coils + glue
14	32/M	Active bleeding from surgical arterial injury	Branch of right colic artery	3	1	None
15 ^b^	79/M	False aneurysm	Superior polar artery of right kidney	3	1	Glue
16 ^a^	64/F	Haemoptysis	Main right bronchial artery	3/5	2	Glue
17 ^a^	70/M	False aneurysm	Gastroduodenal artery	3/5	2	Coils + glue
18	76/F	Haemoptysis	Branch of right Fowler lobe artery	5	1	None
19 ^b^	79/M	False aneurysm	Gastroduodenal artery	5	1	None
20	64/F	False aneurysm	Superior branches of right renal artery	5	6	Coils + glue
21	63/M	Chemo. port catheter	Gastroduodenal artery	5	1	Coils
22	73/M	Left gastric artery aneurysm, incipient rupture	Splenic artery	5	1	Covered stent
23 ^b^	85/M	False aneurysm	Gastroduodenal artery	5	1	Coils
24 ^b^	69/M	False aneurysm	Gastroduodenal artery	5	2	Coils + glue
25	70/M	Chemo. port catheter	Gastroduodenal artery	5	2	None
26	75/M	False aneurysm	Gastroduodenal artery	5	2	Coils + glue
27	74/M	Haemoptysis	Costo-bronchial trunk	5	1	Coils + glue
28	61/F	False aneurysm	Right superior gluteal artery	5	1	Glue
29	75/F	CKD, AV fistula dysfunction	Collateral of basilic vein	5	1	Coils
30	22/M	Dissection with rupture after traffic accident	Right renal artery	5	1	Glue
31	50/F	Common hepatic artery aneurysm	Left hepatic artery	5	1	Coils + covered stent
32	43/M	Subcutaneous haemangioma (AV fistula)	Branch of external carotid artery	5	1	None
33	84/F	CKD, AV fistula dysfunction	Collateral of the cephalic vein	9	1	None

^a^ Technical failure of MVP embolisation (n = 3), ^b^ recurrent bleeding requiring a second embolisation procedure (n = 4), CKD: chronic kidney disease; AV: arterio-venous; AVM: arterio-venous malformation; M: male; F: female.

**Table 3 biomedicines-11-02172-t003:** Details on the nine indications for the 22 emergent procedures used to stop bleeding.

Indications	Number of Patients (Total = 22)
False aneurysm of the pancreaticoduodenal artery	2
False aneurysm of a branch of the gluteal artery	3
False aneurysm of an artery feeding a pancreatic pseudocyst	1
Gastroduodenal artery, duodenal bulb ulcer	6
Haemoptysis from branch of bronchial artery	4
False aneurysm of a renal artery branch	3
Active bleeding from a colic artery branch injured surgically	1
Fissured aneurysm of the left gastric artery	1
Rupture of renal artery dissection due to a traffic accident	1

## Data Availability

All the study data are reported in this article.

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
