# Peer review of "Embolisation Using Microvascular Plugs for Peripheral Applications: Technical Results and Mid-Term Outcomes"

_biomedicines, 2023, doi:10.3390/biomedicines11082172_

Round 1

Reviewer 1 Report

The manuscrip is interesting and it is quite well written. However, I have some comments:

1-Abstract. The objective of this retrospective observational singlecentre study was to assess MVP embolisation procedures performed at a university hospital. The 33 patients who underwent MVP embolisation in 2021 were included (mean age, 64; 24 males and 9 females). The primary endpoint was technical success defined as full first-attempt occlusion with one or more MVPs, documented on the end-of-procedure angiogram. In all, 51 MVPs were used, with other agents in 23 cases (usually coils and/or glue); 22 procedures were emergent. Please, clarify the study sample size.

2- Abstract. The technical success rate was thus 90.9%. No patient experienced MVP migration or other major complications. Five patients had recurrent clinical symptoms; in four cases, the cause was collateral development and in one incomplete initial embolisation. No instances of recanalisation were documented during the short follow-up of 12 months. At our tertiary-level centre, MVP was both effective and safe. Interventional radiologists should be conversant with the techniques and indications of MVP embolisation. Please, ameliorate the description of results. Add, for example, the most important statistically significant values to support the conclusions.

3- INTRODUCTION Since the first report of embolisation in 1972, interventional radiology has steadily gained in importance as a therapeutic tool and field of technological innovation (1,2). Embolisation done to treat lesions or stop bleeding seeks to occlude one or more blood vessels (3). The many available embolisation agents are categorized as resorbable (e.g., gelatine and beads) or permanent (4). The latter include particles (non-resorbable beads), liquids (glues and copolymers), and mechanical devices (coils and plugs) (5). Please, improve the background.

4- INTRODUCTION. The objective of this study was to describe the clinical effectiveness of MVP embolisation used alone or combined with other materials. We considered both scheduled and emergent procedures. Please, improve the description of study aim and underline the novelty of the work.

5- Statistical Analysis Quantitative variables were described as mean±SD if normally distributed and as median [interquartile range] otherwise. Categorical variables were described as number (%). No group comparisons were performed for this study. Ameliorate the description of statistical tests.

6- Figure 2. 60-year-old man (#1) with diffuse metastatic disease from colonic cancer unresponsive to chemotherapy requiring intensification by chemotherapy port-catheter implantation. (a,b) MVP-3Q implantation to occlude the left gastric artery before insertion of the chemotherapy catheter. (c,d) Catheter insertion in the gastroduodenal artery, occlusion of the gastroduodenal artery, and stabilisation of the catheter by implantation of an MVP-3Q. (e,f) Computed tomography after arterial chemotherapy administration. Great figure! Please, improve the legend.

7- CONCLUSION Distal MVP embolisation, usually in combination with other embolic agents, was effective in our study. There were no instances of MVP migration or other major complications. The best indications for using MVP to occlude arteries may be distal arterial bleeding with haemodynamic instability requiring faster haemostasis than expected with coils alone. The MVP can be used in combination with other materials to expedite haemostasis. Interventional radiologists should be conversant with the indications and use of the MVP. Please, underline the novelty of the study and the possible clinical implications.

Author Response

Reviewer #1 comments’ response

The manuscript is interesting and it is quite well written. However, I have some comments:

1-Abstract. The objective of this retrospective observational single-centre study was to assess MVP embolisation procedures performed at a university hospital. The 33 patients who underwent MVP embolisation in 2021 were included (mean age, 64; 24 males and 9 females). The primary endpoint was technical success defined as full first-attempt occlusion with one or more MVPs, documented on the end-of-procedure angiogram. In all, 51 MVPs were used, with other agents in 23 cases (usually coils and/or glue); 22 procedures were emergent. Please, clarify the study sample size.

Reply : Thank you very much for your comment. It has been clarified in the abstract session as suggested.

2- Abstract. The technical success rate was thus 90.9%. No patient experienced MVP migration or other major complications. Five patients had recurrent clinical symptoms; in four cases, the cause was collateral development and in one incomplete initial embolisation. No instances of recanalisation were documented during the short follow-up of 12 months. At our tertiary-level centre, MVP was both effective and safe. Interventional radiologists should be conversant with the techniques and indications of MVP embolisation.

Please, ameliorate the description of results. Add, for example, the most important statistically significant values to support the conclusions.

Reply : Thank you very much for your comment. The abstract has been improved as suggested. However, there is no specific statistical analysis in this paper which is only descriptive. Consequently, the description of the results has been kept to a maximum understanding.

3- INTRODUCTION Since the first report of embolisation in 1972, interventional radiology has steadily gained in importance as a therapeutic tool and field of technological innovation (1,2). Embolisation done to treat lesions or stop bleeding seeks to occlude one or more blood vessels (3). The many available embolisation agents are categorized as resorbable (e.g., gelatine and beads) or permanent (4). The latter include particles (non-resorbable beads), liquids (glues and copolymers), and mechanical devices (coils and plugs) (5). Please, improve the background.

Reply : Thank you very much for your comment. The background has been improved as suggested in the text.

4- INTRODUCTION. The objective of this study was to describe the clinical effectiveness of MVP embolisation used alone or combined with other materials. We considered both scheduled and emergent procedures. Please, improve the description of study aim and underline the novelty of the work.

Reply : Thank you very much for your comment. This part has been improved as suggested in the text.

5- Statistical Analysis Quantitative variables were described as mean±SD if normally distributed and as median [interquartile range] otherwise. Categorical variables were described as number (%). No group comparisons were performed for this study. Ameliorate the description of statistical tests.

Reply : Thank you very much for your comment. As suggested, this part has been improved. Howefer, no specific statistical tests were performed since the study was purely descriptive as mentionned.

6- Figure 2. 60-year-old man (#1) with diffuse metastatic disease from colonic cancer unresponsive to chemotherapy requiring intensification by chemotherapy port-catheter implantation. (a,b) MVP-3Q implantation to occlude the left gastric artery before insertion of the chemotherapy catheter. (c,d) Catheter insertion in the gastroduodenal artery, occlusion of the gastroduodenal artery, and stabilisation of the catheter by implantation of an MVP-3Q. (e,f) Computed tomography after arterial chemotherapy administration. Great figure! Please, improve the legend.

Reply : Thank you very much for your comment. Legend has been improved as suggested. Figure has been renumbered as Figure 3.

7- CONCLUSION Distal MVP embolisation, usually in combination with other embolic agents, was effective in our study. There were no instances of MVP migration or other major complications. The best indications for using MVP to occlude arteries may be distal arterial bleeding with haemodynamic instability requiring faster haemostasis than expected with coils alone. The MVP can be used in combination with other materials to expedite haemostasis. Interventional radiologists should be conversant with the indications and use of the MVP. Please, underline the novelty of the study and the possible clinical implications.

Reply : Thank you very much for your comment. The novelty of the study has been underlined as suggested in this section. Furthermore, the possible clinical implications have been emphasized.

Reviewer 2 Report

The work concerns embolisation. As the authors droughtily note, there are not many parcs in this field. The authors have very precisely defined the treatment indications in the form of a table which makes the work very easy to read. The patient group is large enough. The authors demonstrated the usefulness of the technique used, above all its efficiency. The literature counts 23 items, but in the case of the topic undertaken, such a small number is justified.

Four sizes are available- please specify

The catheter was flushed with saline- barren?

Please post a photo of the device

Author Response

Reviewer #2 comments’ response

The work concerns embolisation. As the authors droughtily note, there are not many parcs in this field. The authors have very precisely defined the treatment indications in the form of a table which makes the work very easy to read. The patient group is large enough. The authors demonstrated the usefulness of the technique used, above all its efficiency. The literature counts 23 items, but in the case of the topic undertaken, such a small number is justified.

Reply : Thank you very much for your comment. Nothing to add.

Four sizes are available- please specify

Reply : Thank you very much for your comment. It has been specified in the embolization technique paragraph of the materials and methods section.

The catheter was flushed with saline- barren?

Reply : Thank you very much for your comment. Yes, the catheter was flushed before MVP insertion. It has been clarified in the embolization technique paragraph  of the materials and methods section.

Please post a photo of the device

Reply : Thank you very much for your comment. A picture of the device (Figure 1) has been added to the manuscript as suggested. Figures have been renumbered consequently.

Round 2

Reviewer 1 Report

No further comments